# Long-Term Outcome of Neonatal Seizure with *PACS2* Mutation: Case Series and Literature Review

**DOI:** 10.3390/children10040621

**Published:** 2023-03-26

**Authors:** I-Jun Chou, Ju-Yin Hou, Wen-Lang Fan, Meng-Han Tsai, Kuang-Lin Lin

**Affiliations:** 1Division of Pediatric Neurology, Department of Pediatrics, Linkou Chang Gung Children’s Hospital and Linkou Chang Gung Memorial Hospital, Taoyuan City 333, Taiwan; 2School of Medicine, College of Medicine, Chang Gung University, Taoyuan City 333, Taiwan; 3Department of Medical Research, Kaohsiung Chang Gung Memorial Hospital, Kaohsiung City 833, Taiwan; 4Department of Neurology, Kaohsiung Chang Gung Memorial Hospital, Kaohsiung City 833, Taiwan

**Keywords:** developmental and intellectual disabilities, early infantile epileptic encephalopathy (EIEE), epilepsy, neonatal seizure, *PACS2*

## Abstract

*Phosphofurin Acidic Cluster Sorting Protein 2 (PACS2)*-related early infantile developmental and epileptic encephalopathy (EIDEE) is a rare neurodevelopmental disorder. EIDEE is characterized by seizures that begin during the first three months of life and are accompanied by developmental impairment over time. In this article, we present three patients with EIDEE who experienced neonatal-onset seizures that developed into intractable seizures during infancy. Whole exome sequencing revealed a de novo heterozygous missense variant in all three patients in the p.Glu209Lys variant of the *PACS2* gene. We conducted a literature review and found 29 cases to characterize the seizure patterns, neuroimaging features, the usage of anticonvulsants, and the clinical neurodevelopmental outcomes of *PACS2*-related EIDEE. The seizures were characterized by brief, recurring tonic seizures in the upper limbs, sometimes accompanied by autonomic features. Neuroimaging abnormalities were observed in the posterior fossa region, including mega cisterna magna, cerebellar dysplasia, and vermian hypoplasia. The long-term prognosis ranges from low–average intelligence to severe developmental retardation, emphasizing the importance of early recognition and accurate diagnosis by pediatric neurologists to provide personalized patient management.

## 1. Introduction

Early infantile developmental and epileptic encephalopathy (EIDEE) is a rare and severe form of epilepsy that typically presents during the first 3 months of life, with developmental impairment that becomes evident over time [1]. The incidence of seizures in neonates and infants is estimated to be 7/100,000 live births [2,3]. Common features of EIDEE include frequent seizures, developmental delays, and brain abnormalities. Symptoms and disease progression can vary greatly. Genetic mutations are the cause of many forms of EIDEE, and a genetic test may confirm the diagnosis.

*Phosphofurin Acidic Cluster Sorting Protein 2 (PACS2)*-related EIDEE is a specific type of EIDEE caused by mutations in the *PACS2* gene. *PACS2* is a multifunctional protein that plays a vital role in maintaining the balance and proper functioning of the mitochondria, endoplasmic reticulum (ER), and lysosomes. It is also a crucial regulator of mitochondria-associated membranes [4]. Alterations in the autoregulatory domain of *PACS2* impair the ability of *PACS2* to control the interaction between its cargo(furin)-binding regions and client proteins, potentially leading to disruptions in cellular function [5,6]. The relationship between *PACS2* mutations and seizures is not fully understood. However, *PACS2* may be involved in ion channel regulation [7] such that ion channel functions are disrupted, potentially leading to seizures. Disruptions in ion channel functions could potentially lead to seizures. For example, paroxysmal depolarization shift (PDS) is a hallmark of epileptiform activities at the cellular level in both lower invertebrates and higher mammals, including humans. Investigating simpler invertebrate model systems such as leech Retzius nerve cells can help us achieve an understanding of the complexity of K+ channels and their roles in epilepsy. Research highlights the non-synaptic nature of PDS, which arises from the suppression of calcium-activated K+ channel functions due to a voltage-gated calcium channel blockade and the unmasking of the persistent sodium current, leading to Na+-dependent PDS in leech Retzius nerve cells [8,9]. Further research is needed to elucidate the connection between PACS2 mutations, ion channel regulation, and the development of seizures.

The prevalence of pediatric epilepsy, a neurological disorder, is high, and some forms of epilepsy have been linked to genetic mutations. It is important for pediatric neurologists to recognize specific clinical presentations associated with such disorders, including *PACS2*-related EIDEE, to ensure timely identification and appropriate treatment and consultation. Here, we present three cases of *PACS2*-related EIDEE, first reported in Taiwan. To characterize the clinical characteristics and long-term outcomes of *PACS2*-related EIDEE, we further identified 29 cases through a literature review.

## 2. Materials and Methods

The study was approved by the Institutional Review Board of Chang Gung Memorial Hospital (IRB No. 201701524A3C501 and IRB No. 202101571B0), and informed consent was obtained from the patients’ parents. Informed consent was also obtained from the parents of each patient for the use of the photos.

### 2.1. Study Objective

We aimed to identify seizure patterns, a treatment used for seizures, the outcome of seizures, dysmorphism, neuroimaging features, physical abnormalities, and neurodevelopmental outcomes.

### 2.2. Study Design

The enrollment criteria were consecutive cases with epilepsy identified at the Linkou branch of the Chang Gung Memorial Hospital between 2010 and 2022. All patients had genetic tests, including whole exome sequencing, variant filtering, and prioritization, and a heterozygous missense variant, *PACS2*:c.625G>A(p.Glu209Lys) (GenBank: NM_001100913.3), was revealed.

Genomic DNA was isolated from peripheral venous leukocytes. Whole exome sequencing was performed using two different pipelines. Case 1 was captured using Agilent Clinical Research Exome V2 and sequenced using the Illumina NovaSeq6000 platform (Illumina, San Diego, CA, USA). The obtained sequences were aligned with the reference human genome (hg19) using the Burrows–Wheeler Aligner, and variants were called using the Genome Analysis ToolKit. Variants were annotated using wANNOVAR and filtered to obtain variants located in exons or close to splice sites and those not present in the 1000 Genomes Project (http://www.1000 genomes.org, accessed on 27 June 2022), the Exome Aggregation Consortium database, or the Genome Aggregation Database (https://gnomad.broadinstitute.org, accessed on 27 June 2022). The integrated genome viewer was used to visualize the reads and variants. Direct Sanger sequencing was performed to verify the genetic variants detected by whole exome sequencing. Pathogenicity was classified according to the American College of Medical Genetics and Genomics guidelines. Cases 2 and 3 were recruited from the international Epi25 collaborative project; the methods were detailed previously [10]. The de novo occurrence of this missense variant was confirmed by Sanger sequencing.

Comprehensive clinical data, including demographic information, laboratory results, imaging studies, electrophysiological evaluations (consisting of non-sedated video electroencephalography for a minimum of 24 h during active seizures in infancy), and the hospital courses were retrospectively obtained from the patients’ electronic medical records in 2022. Two experienced pediatric neurologists (I.J.C. and K.L.L.) evaluated the clinical presentations and assessed the seizure patterns, dysmorphism, and neuroimaging features.

Prior to the genetic study, all patients underwent metabolic and neuroimaging surveys. The metabolic survey included tandem mass spectrometry, urinary organic acid levels, serum lactate and pyruvate levels, and routine blood biochemistry. Neurological imaging included neonatal brain ultrasonography through the anterior fontanelle and a contrast-enhanced brain MRI at 1.5 Tesla (T), performed during the study period. Patients were sedated and positioned comfortably within the MRI scanner (GE HealthCare), and a standard head coil was employed to ensure adequate signal reception. T1-, T2-, and fluid-attenuated inversion recovery (FLAIR)-weighted sequences were performed to capture detailed structural information. The acquired images were then assessed by senior pediatric neuroradiologists.

### 2.3. Intervention

The intervention for seizure control and its efficacy was reviewed.

### 2.4. Outcome Measure

The outcome measures included seizures and clinical neurodevelopmental outcomes at the last follow-up. The seizure outcome was defined as controlled or uncontrolled, according to seizure frequency. Neurodevelopmental outcome measures include motor, speech, and learning abilities and neuropsychiatric conditions.

### 2.5. Literature Review

A literature review was conducted using the PubMed database, with the language restriction set to English, Chinese, and Japanese. The search strategy employed the specific keywords seizure, epilepsy, *PACS2*, and developmental delay. The identified articles were critically appraised for relevance, and their clinical, genetic, and neuroimaging findings were examined. The assessment includes seizure patterns, the final usage of anticonvulsants, seizure outcomes, dysmorphism, neuroimaging features, and clinical neurodevelopmental outcomes. We presented descriptive statistics. Quantitative data are presented as medians with interquartile ranges (IQRs) and categorical data as frequencies with percentages.

## 3. Results

### 3.1. Patients

Three cases with a de novo heterozygous missense variant, *PACS2*:c.625G>A (p.Glu209Lys), were enrolled in the case series.

#### 3.1.1. Case 1

A male infant born at 39 weeks of gestation with Apgar scores of 9 and 10 at 1 and 5 min, respectively, was diagnosed with right hydronephrosis at birth, which was later determined to be a horseshoe kidney. At 2 weeks of age, the patient had his first seizure, which was treated with oral diazepam at a local hospital for 1 month. At 3 months, he presented with recurrent clusters of generalized tonic–clonic seizures (GTCs) during a feverish episode of pneumonia. A gross motor delay was noted at admission, and an EEG revealed focal cortical dysfunction over the right frontal area. Despite treatment with phenobarbital and oxcarbazepine, the patient continued to experience recurrent generalized seizures. A brain ultrasonography and MRI revealed prominent extracerebral cerebrospinal fluid spaces and mega cisterna magna (Figure 1A). The results of metabolic analyses, including lactate, pyruvate, and total and free carnitine levels, blood tandem mass spectrometry, and urine gas chromatography–mass spectrometry, were within normal limits. His cardiac, lumbosacral spinal, and abdominal ultrasound findings were also unremarkable. At 6 months of age, the patient presented with repetitive spasms in the left face, left arm, and hand as well as postural tonic spasms that occurred in clusters. A long-term video EEG showed interictal epileptic discharges at F3, C3, Cz, and C4. The patient was administered intravenous pyridoxal phosphate of up to 240 mg/day with a successful response, but the seizures recurred after he was shifted to oral inactive vitamin B6. Although a ketogenic diet was initiated, it was later discontinued due to dietary intolerance. Levetiracetam was then added and was effective at controlling the seizures. Until 23 months of age, the patient showed signs of motor and speech delays, in addition to synophrys and wide-spaced teeth. A follow-up brain MRI at 3 years of age was unremarkable. At 5 years of age, the patient had been seizure-free for 2 years while on vitamin B6 and levetiracetam. The final assessment revealed hypotonia and a speech delay.

#### 3.1.2. Case 2

A female infant was delivered via Cesarean section at 36 weeks of gestation due to premature rupture of the membranes. At approximately 1 month of age, she had her first seizure, characterized by grasping and a tonic trunk posture. She was seen at a local clinic which recommended observation. Four months later, the seizures recurred and presented as tonic movements in all four limbs, facial flushing, cyanosis of the lips with drooling, and an upward gaze lasting for 1 min, followed by a period of postictal lethargy for 30 min. The frequency of the seizures increased over time, evolving into tonic–clonic seizures. The initial interictal EEG showed slow waves in both hemispheres, with focal spikes later identified over the bilateral centrotemporal areas. The infant was treated with intravenous pyridoxal phosphate at 40 mg/day, which was effective during her hospitalization, but the seizures resumed after she was shifted to oral inactive vitamin B6. Therefore, oral pyridoxal phosphate was reused. The initial MRI showed vermian hypoplasia and a mega cisterna magna (Figure 1B). The results of the metabolic analyses were within normal limits. Facial dysmorphism with hypertelorism was found in infancy. Her chromosome karyotype was normal (46, XX). Oxcarbazepine was added to the treatment regimen for seizures occurring in conjunction with urinary tract infections. After 3 years of being seizure-free, recurrent seizures with GTCs during sleep and generalized spikes on EEG prompted a shift in anticonvulsant therapy to levetiracetam and valproic acid. A follow-up MRI at 7 years of age showed persistent mega cisterna magna, vermian hypoplasia, mild prominence of the cisterns and sulci of the bilateral cerebral hemispheres, and left mesial temporal sclerosis. Over several years of follow-up, interictal EEGs revealed activity over the right centrotemporal area, which normalized at 10 years of age under the administration of valproic acid and vitamin B6. The patient had a history of gross motor and speech delays during early childhood and was later diagnosed with attention deficit hyperactivity disorder, which was treated with methylphenidate. The patient also had a history of repetitive urinary tract infections before 2 years of age and was diagnosed with right hydronephrosis and bilateral vesicoureteral reflux. An atrial septal defect was detected on cardiac ultrasonography. Figure 2A shows facial dysmorphism at 10 years.

#### 3.1.3. Case 3

A female was born at 38 weeks of gestation via Cesarean delivery due to premature rupture of the membranes. Her birth weight was 3120 g, and her Apgar score was 7 at 1 min and 9 at 5 min. Upon delivery, poor sucking and feeding were observed. The patient had her first seizure at 2 weeks of age, characterized by screaming, general muscle tightening, and a facial flush lasting approximately 1 min. Due to an increased seizure frequency of up to 10 times/day, she was referred to our center at 3 weeks of age. Upon examination at our hospital, focal seizures were identified that presented with symptoms such as upward gaze, jerking of the left limbs, head turning to the right, eye deviation to the left, dilated pupils without light reflex, and desaturation that lasted for approximately 1 min. Despite initial treatment with phenobarbital, her seizures occurred 20 times/day, and further seizures with yawning or hiccuping followed by bicycling movements of the legs were observed, leading to the patient’s transfer to the neonatal intensive care unit. Continued seizures with grasping and facial flushing accompanied by desaturation were identified. An initial video 24-h EEG revealed diffuse cortical dysfunction and multifocal epileptiform discharges over the O2, T3, C3, and O1 electrodes. Levetiracetam and intravenous pyridoxal phosphate at 40 mg/day were added to the patient’s treatment regimen. A brain MRI revealed mega cisterna magna (Figure 1C). The results of metabolic analyses, including lactate, pyruvate, and total and free-carnitine levels, blood tandem mass spectrometry, and urine gas chromatography–mass spectrometry, were all within normal limits. The patient’s chromosome karyotype was normal (46, XX). Over the following years, she had multiple hospital admissions for fever with GTCs and required intubation due to pneumonia. The findings of a follow-up EEG were similar to those of the initial EEG. The patient was diagnosed with a global developmental delay, hypotonia, horizontal nystagmus, feeding difficulties, gastroesophageal reflux disease, failure to thrive, and microcephaly. A combination of phenobarbital, vigabatrin, and clonazepam was administered for several years, reducing seizure frequency to once or twice monthly in subsequent years. Additionally, an atrial septal defect was detected via cardiac ultrasonography. Figure 2B shows facial dysmorphism at 6 years.

### 3.2. Literature Review

#### 3.2.1. Demographics of 32 Cases

Based on previous reports [6,11,12,13,14,15,16,17,18] and the three patients described herein, 32 cases of PACS2-related EIDEE (females 63% and males 37%) are known thus far. In 31 patients, seizure onset occurred during the neonatal stage; in the remaining patient, it was first reported at 2 months. The average age of seizure onset was 10 (median, 6.5 days; IQR, 3–14) days. The average age at the final follow-up was 7 years (range, 5 months to 37 years; median, 5 years; IQR, 2–9.5 years). Most patients had the heterozygous missense variant *PACS2*:c.625G>A (p.Glu209Lys), but one patient had the heterozygous missense variant *PACS2*:c.631G>A (p.Glu211Lys). All but one patient had a de novo mutation. One patient inherited the p.Glu209Lys variant from the mother, who also had a characteristic clinical phenotype [16].

#### 3.2.2. Seizure Patterns, Intervention, and Outcomes

The 32 patients with PACS2-related EIDEE identified thus far presented with neonatal onset seizures which were initially controlled by a single anticonvulsant medication but evolved into intractable seizures within several months. Many of the patients were admitted to the hospital due to the occurrence of seizures in conjunction with infectious fevers. Table 1 shows the seizure patterns, treatment, and outcome for each patient. Pyridoxine phosphate was initially administered in some cases. Although pyridoxine phosphate was partially effective, genetic testing for pyridoxamine 5′-phosphate oxidase (PNPO) yielded negative results in four of the patients (1, 2, 3, and 31) [17]. Pyridoxamine 5′-phosphate oxidase deficiency is a newly recognized form of autosomal recessive neonatal epileptic encephalopathy (MIM#610090), in which the seizures can be controlled by the active form of pyridoxal 5′-phosphate but not always by pyridoxine (B6). By the age of 2 years, the seizures had been managed in the majority of the patients with antiseizure medications. Half of the patients were effectively treated with either valproate or levetiracetam; one-third of patients showed improvement with phenobarbital, and one-fourth with carbamazepine. Combination therapy was often necessary.

At the final follow-up, out of 17 patients aged 5 years or older, 5 (29%) had discontinued anticonvulsant use, and 2 adults had been seizure-free for over a decade. One patient (patient 22) developed Lennox–Gastaut syndrome after a period of seizure control with anticonvulsants [12].

#### 3.2.3. Dysmorphism

Physical dysmorphism may not be present during the neonatal stage, but thinness of the upper lip and downturned mouth corners are commonly reported. Other features noted among the patients included synophrys, high-arched eyebrows, hypertelorism, downward-slanting palpebral fissures, and widely spaced teeth. A horseshoe-shaped kidney and bilateral vesicoureteral reflux were observed in our patients (patients 1 and 3, respectively), but neither of these findings was documented in the literature.

#### 3.2.4. Neuroimaging Features

Brain MRIs showed characteristic posterior fossa abnormalities, including mega cisterna magna, in 43% (13/30) of patients; cerebellar dysgenesis (e.g., foliar distortion) was described in 40% (12/30), vermian hypoplasia in 23% (7/30), and a retro-cerebellar cyst in one patient. Negative neuroimaging results were reported in 20% (6/30) of the patients. While most patients had a normal head circumference, two (6%) had microcephaly, which may have been due to persistent feeding problems (patient 3) or intractable seizures (patients 3 and 5) [11].

#### 3.2.5. Clinical Neurodevelopmental Outcomes

Delays in gross motor development and walking were common features. Most patients were able to walk at 3 years of age. A wide-based gait was noted in one patient (patient 32) that persisted into adulthood [18]. Speech delay was pervasive, and autism spectrum disorders were diagnosed in 18% (4/22) of patients. Moderate to severe developmental delays in childhood were reported in 25% of patients. At school age, the patients had at least a low-to-average IQ and had graduated from mainstream schools or colleges. One patient (patient 30) had learning disabilities in adolescence [16], and one (patient 32) had a college degree [18].

## 4. Discussion

*PACS2*-related EIDEE is distinguished by recurrent focal seizures and autonomic dysfunction, with an early onset in infancy. Notable facial anomalies are also present. A brain MRI may demonstrate aberrations in the posterior cranial fossa. Thus, it is crucial that pediatric neurologists recognize these hallmark features and order genetic testing for *PACS2* mutations to promptly diagnose this genetic condition and provide individualized management. The first seizure typically occurs within the first month of life. Among the patients included in this study and review, seizure onset occurred during the first month in all but one, and in half of the patients, the seizures occurred during the first week of life. Focal spasms or GTCs with drug-resistant features may emerge during the following months in addition to autonomic signs such as apnea, cyanosis, and vocalizations. After treatment with a minimum of two anticonvulsant medications, with partial efficacy achieved with pyridoxine, seizure control was successful in half of the patients by age 2–5 years. Information on the long-term outcomes of epilepsy in patients older than 6 years is still limited, with reported results including complete seizure freedom, seizure control through antiseizure medications, intractable epilepsy, and the development of Lennox–Gastaut syndrome. Microcephaly has been linked to severe developmental delays.

*PACS2*, a multifunctional sorting protein, regulates the interplay between the ER and mitochondria, maintaining ER homeostasis [4]. The link between epilepsy and PACS2 remains unclear, but may involve mutations affecting ion channel function, leading to seizures. *PACS2* gene expression is associated with ion channel binding sites, impacting epileptiform activities [7]. Various mechanisms, such as Kv channel mutations, non-synaptic origin of epileptiform activities, and ion channel blocks, play crucial roles in seizures. Studies demonstrated that Ni(2+)-induced epileptic activity is suppressed by ethanol and magnesium, with Kv1 channels differentially regulating action potentials in etv1 and glt pyramidal neurons. These findings highlight diverse ion channel interactions in epileptiform activity regulation at the cellular level [19,20,21].

The efficacy of vitamin B6 as a therapeutic approach for seizures in *PACS2*-related EIDEE may be due to its ability to protect against mitochondrial dysfunction [22]. Our three cases demonstrated a transient, significant response to the pyridoxal phosphate treatment which may be explained by the mitochondrial protection effects. However, we should conduct further function tests to clarify the mechanisms. Furthermore, *PACS2*-depleted cells have a heightened susceptibility to apoptosis, particularly under stressful conditions [23], which may explain the correlation between fever and seizures, as observed in infants.

A neuroimaging hallmark of *PACS2*-related EIDEE is the presence of posterior fossa abnormalities. This may include cerebellar dysplasia, which is characterized by cerebellar gyri irregularities [24] and requires meticulous evaluation by a radiologist. The cerebellar cortex is much denser than the cerebral cortex and accounts for a significant portion of the neocortical surface area [25]. Vermian hypoplasia can be assessed through the tegmento–vermian angle. At the fetal and neonatal stages, the normal tegmento–vermian angle is typically <18° [26]. An angle of <30° combined with a normal cerebellar morphology indicates no vermian hypoplasia, and an angle of ≥30° indicates the presence of vermian hypoplasia [27]. An elevated tegmento–vermian angle > 40–45° is commonly associated with classic Dandy–Walker malformation [26]. Infants with isolated inferior vermian hypoplasia may exhibit normal development at 6 years. In a prospective study of 20 patients, roughly 10% had developmental delays or behavioral issues [28]. The vermis receives sensory inputs from the head, trunk, and limbs as well as vestibular, auditory, and visual inputs, and it outputs control signals to the cortex and brainstem via the fastigial nucleus [29]. Mega cisterna magna, defined as a projection of >10 mm from the posterior aspect of the vermis to the interior aspect of the skull vault in fetuses or neonates, is believed to stem from defects in the posterior membranous region during embryogenesis [30]. Given that the cerebellum continues to develop during the first postnatal year [24], neuroimaging evaluations may need to be repeated. The impact of the isolated mega cisterna magna has not been investigated extensively, although one study found suboptimal performance on memory evaluations, such as the Rey Auditory Verbal Learning Test, and in verbal fluency [31].

The developmental consequences in infants diagnosed with *PACS2*-related EIDEE, regardless of the presence or absence of posterior fossa abnormalities, include low-to-average IQ, cognitive impairment of varying degrees, developmental delays in motor skills and equilibrium, delays in language development, hypotonia, oculomotor abnormalities, and behavioral issues.

The present investigation is subject to certain limitations that need to be acknowledged. Firstly, the inclusion criteria solely comprised individuals diagnosed with epilepsy at the Linkou branch of Chang Gung Memorial Hospital from 2010 to 2022, which may not be a representative sample of the entire population. Consequently, the generalizability of the research findings could be restricted. Moreover, the study design was retrospective, and the data were obtained from the electronic medical records of patients, which might have incomplete or erroneous documentation. Additionally, the sample size of the investigation was limited, with only three cases enrolled. Furthermore, the literature review was confined to the English, Chinese, and Japanese languages, which may have constrained the scope of the search results. Future research should conduct a study with a larger and more inclusive sample, utilizing multiple data sources. The literature review should also expand to include publications in other languages and databases, providing a more comprehensive understanding of *PACS2*-related EIDEE for clinical practice.

## 5. Conclusions

*PACS2*-related EIDEE is a rare disorder characterized by neonatal-onset seizures, facial dysmorphism, and speech delay, with brain posterior fossa abnormalities seen in the majority of patients. The outcome is variable. Genetic testing for *PACS2* mutations is crucial for early diagnosing of *PACS2*-related EIDEE and its personalized management.

## Figures and Tables

**Figure 1 children-10-00621-f001:**
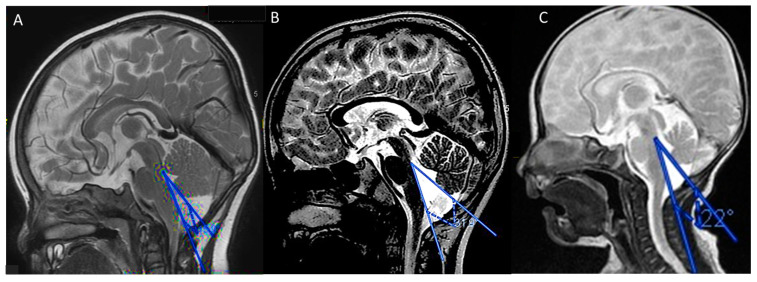
MRI image features (demonstrated by T2-weighted imaging). (**A**) Case 1 with mega cisterna magna. (**B**) Case 2 with vermian hypoplasia (with a tegmento-vermian angle at 32 degrees). (**C**) Case 3 with mega cisterna magna. The tegmento-vermian angle (illustrated using blue-colored lines) is the angle between the brain stem’s dorsal side and the ventral part of the vermis. An angle of ≥30° indicates the presence of vermian hypoplasia.

**Figure 2 children-10-00621-f002:**
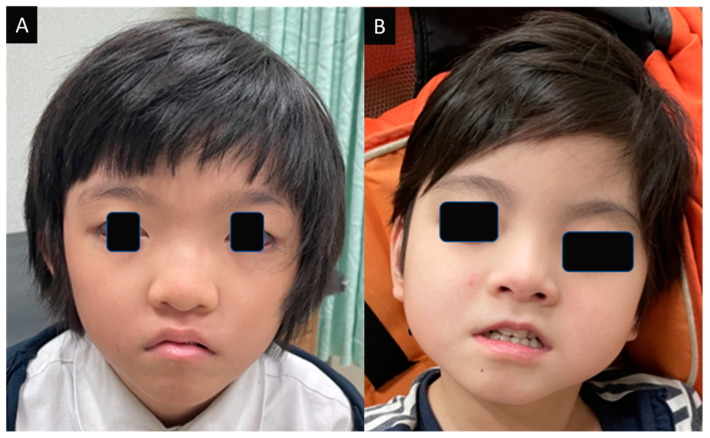
Facial features. Both Case 2 (**A**) and Case 3 (**B**) had facial dysmorphism, including synophrys (meeting of the medial light, sparse eyebrows in the midline), hypertelorism, downward-slanting palpebral fissures, a thin upper lip vermilion, and down-turned corners of the mouth (photos were captured by mobile iPhone 12).

**Table 1 children-10-00621-t001:** Summary of the clinical phenotypic features in 32 patients with PACS2-related early infantile developmental and epileptic encephalopathy (EIDEE).

Case	Sex	Age of Onset	Seizure Types	MRI	WES (*PACS2*, Heterozygous Missense Variant)	Inheritance	Age at Last Follow Up	AEDs at Last Follow-Up	Seizure Frequency	Development Outcome	Ref.
1	M	2 w	GTCs, focal (upper limb)	MCM	c.625G>A (p.Glu209Lys)	de novo	5 y	B6, LEV	controlled	hypotonia, speech delay	see Case 1
2	F	1 m	tonic, autonomic dysfunction	VH, MCM	c.625G>A (p.Glu209Lys)	de novo	10 y	B6, VPA	controlled	ADHD	see Case 2
3	F	2 w	tonic, autonomic dysfunction, focal (head-turning, upper limbs, left leg, cycling), GTCs, vocalizations	MCM; Other: microcephaly	c.625G>A (p.Glu209Lys)	de novo	6 y	BP, VGB, CLN	uncontrolled	severe developmental delay, severe MR, hypotonia	see Case 3
4	M	10 d	focal, tonic	negative	c.625G>A (p.Glu209Lys)	de novo	5 y	B6, LEV, VPA	controlled	moderate developmental delay	Hu, C. et al. [11]
5	F	1 m	focal, tonic	microcephaly	c.625G>A (p.Glu209Lys)	de novo	2 y	B6, LEV, TPM, BP, VPA, KD	controlled	severe developmental delay	Hu, C. et al. [11]
6	F	1 m	focal, tonic	negative	c.625G>A (p.Glu209Lys)	de novo	2 y	B6, LEV, VPA	controlled	moderate developmental delay	Hu, C. et al. [11]
7	F	6 d	focal	CD, MCM	c.625G>A (p.Glu209Lys)	de novo	16 y	CBZ			Olson, H.E. et al. [6]
8	F	4 d	GTCs	CD, VH, MCM	c.625G>A (p.Glu209Lys)	de novo	4 y	PB, VPA	controlled	wide-based gait, sleeping, and behavioral disturbances	Olson, H.E. et al. [6]
9	M	4 d		Other: increased subarachnoid spaces	c.625G>A (p.Glu209Lys)	de novo	15 y	CBZ			Olson, H.E. et al. [6]
10	F	7 d	GTCs	CD, VH, MCM	c.625G>A (p.Glu209Lys)	de novo	8 y	B6, P5P, VPA, PB	controlled	ASD	Olson, H.E. et al. [6]
11	M	2 d	clonic and GTCs	CD, VH, MCM	c.625G>A (p.Glu209Lys)	de novo	1 y 7 m	LEV, PB, CBZ	controlled		Olson, H.E. et al. [6]
12	M	2 d	status epilepticus	negative	c.625G>A (p.Glu209Lys)	de novo	8 y	TPM	controlled	OCD	Olson, H.E. et al. [6]
13	M	2 d	focal, tonic, autonomic dysfunction	VH, other: retrocerebellar cyst	c.625G>A (p.Glu209Lys)	de novo	1 y 4 m	PB			Olson, H.E. et al. [6]
14	F	2 w	focal, tonic	negative	c.625G>A (p.Glu209Lys)	de novo	5 y	AEDs discontinued	AEDs discontinued at 3.5 y	atypical social and behavioral features	Olson, H.E. et al. [6]
15	F	2 d	focal, tonic, myoclonic, GTCs	CD, MCM	c.625G>A (p.Glu209Lys)	de novo	3 y	LEV, PB	controlled	atypical social and behavioral features	Olson, H.E. et al. [6]
16	M	1–2 m	clonic, GTCs	MCM	c.625G>A (p.Glu209Lys)	de novo	7 y	LEV	controlled	ASD	Olson, H.E. et al. [6]
17	M	1 d	focal, GTCs	VH	c.625G>A (p.Glu209Lys)	de novo	12.5 y	VPA		ASD	Olson, H.E. et al. [6]
18	F	3 d	focal, tonic, GTCs, status epilepticus	CD, MCM	c.625G>A (p.Glu209Lys)	de novo	9 m	LEV, PB, OCZ			Olson, H.E. et al. [6]
19	F	2 w	focal, GTCs	CD	c.625G>A (p.Glu209Lys)	de novo	3.5 y	B6, P5P, LEV, VGB, LMT, VPA, CLB			Olson, H.E. et al. [6]
20	F	3 d	tonic, GTCs	CD, MCM; Other: SAH	c.625G>A (p.Glu209Lys)	de novo	5.5 y	B6, PB, LEV, LCS		wide-based gait, selective mutism	Olson, H.E. et al. [6]
21	F	2 w	tonic, head-turning	other: signs of perinatal injury	c.625G>A (p.Glu209Lys)	de novo	2 y	CBZ, CLB	controlled	hypotonia, mild developmental delay	Mizuno, T. et al. [12]
22	F	3 d	focal, GTCs	Normal	c.625G>A (p.Glu209Lys)	de novo	12 y	LMT, VPA, CLN	4 y controlled; 9 y Lennox–Gastaut syndrome; 12 y almost controlled	ASD, severe MR, hypotonia, walking alone	Mizuno, T. et al. [12]
23	F	3 d	tonic	Other: right venous sinus thrombosis	c.625G>A (p.Glu209Lys)	de novo	3 y	AEDs discontinued	AEDs discontinued	normal psychomotor development	Mizuno, T. et al. [12]
24	F	7 d	focal, GTC		c.625G>A (p.Glu209Lys)	de novo	2 y 2 m	VPA	uncontrolled (poor compliance)		Wu, M.J. et al. [13]
25	F	5 d	focal	CD	c.625G>A (p.Glu209Lys)	de novo	5 m	LEV, VPA	controlled		Wu, M.J. et al. [13]
26	F	3 d	tonic		c.625G>A (p.Glu209Lys)	de novo	5 m	LEV, VPA	controlled		Wu, M.J. et al. [13]
27	M	3 d	focal (upper limb)	CD	c.631G>A (p.Glu211Lys)	de novo	7 y	AEDs discontinued	AEDs discontinued	moderate MR (IQ = 47)	Dentici, M.L. et al. [14]
28	F	2 d	tonic (upper limbs), autonomic dysfunction	CD	c.625G>A (p.Glu209Lys)	de novo	2 y 2 m	CBZ	controlled		Terrone, G. et al. [15]
29	M	7 d	focal	VH, MCM	c.625G>A (p.Glu209Lys)	Maternal Inheritance	1 y 7 m	CBZ	uncontrolled	hypotonia, moderate to severe developmental retardation	Cesaroni, E. et al. [16]
30	F	a few weeks	tonic	CT: negative	c.625G>A (p.Glu209Lys)		37 y	seizure free	AEDs discontinued at several years old	learning disability	Cesaroni, E. et al. [16]
31	M	8 d	tonic	MCM	c.625G>A (p.Glu209Lys)	de novo	11 y	seizure free	AEDs discontinued; B6, P5P, folinic acid	mild MR (IQ = 62), ASD, motor and speech delay	Perulli, M. et al. [17]
32	M	2 m	tonic	CD, other: cerebellar progressive atrophy	c.625G>A (p.Glu209Lys)	de novo	23 y	seizure free	AEDs discontinued at 9 y	low average IQ (IQ = 85), hypotonia, wide-based gait	Sakaguchi, Y. et al. [18]

Abbreviations: ADHD—attention-deficit hyperactivity disorder; AEDs—antiepileptic drugs; ASD—autism spectrum disorder; B6—vitamin B6 (pyridoxine); BP— Phenobarbital; CD—cerebellar dysgenesis; CBZ—Carbamazepine; CLB—clobazam; CLN—Clonazepam; CT—computed tomography; d—day; F—female; GTCs —generalized tonic–clonic seizures; IQ—intelligence quotient; KD—ketogenic diet; LCS—Lacosamide; LEV—Levetiracetam; LMT—Lamotrigine; m—month/months; M—male; MCM—mega cisterna magna; MR—mental retardation; MRI—magnetic resonance imaging; OCD—obsessive-compulsive disorder; OCZ— oxcarbazepine; P5P—pyridoxal 5′-phosphate; Ref.—reference; SAH—subarachnoid hemorrhage; TPM—Topiramate; VH—vermian hypoplasia; VGB—Vigabatrin; WES—whole exome sequencing; w—week/weeks; y—year/years.

## Data Availability

The data presented in this study are available upon request from the corresponding author. The data are not publicly available due to patient privacy.

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
