# Peer review of "Long-Term Outcome of Neonatal Seizure with PACS2 Mutation: Case Series and Literature Review"

_children, 2023, doi:10.3390/children10040621_

Round 1

Reviewer 1 Report

 The topic "Neonatal seizure with PACS2 mutation~ long-term outcome and literature review" is an interesting article that the author has communicated, where the author determined that gene-related disorder at an early stage is needed for diagnosis and patient management.

However, there are weaknesses in the article as well.

Line 13-25: I would rather provide the full form of the word when it is used for the first time in the manuscript. Eg; PACS2, EIDEE etc.
Still, the sample size (patient) is low n=3! But since it is a human study and it is understandable the research subject is precious and the greater sample size is difficult to achieve!

The introduction and discussion section currently needs more refinement and elaboration to cover depth research bringing various possible mechanisms into context.

Line 46-47: Please add another paragraph for the origin, propagation, and termination of seizure at the cellular level. This will give a generalized perspective of epilepsy and does not only limit to the introduction of human cases. For details consult:

1)https://medcraveonline.com/medcrave.org/index.php/MOJAP/article/view/19872/pdf

2) http://www.doiserbia.nb.rs/Article.aspx?id=0354-46641004035P#.ZATTYx_MLrc
With this suggested article, you should be able to define Paroxysmal depolarization shift (PDS) a hallmark of epileptiform activities at the cellular level, and how such features are identical in higher mammals and humans at a cellular level.

Line 240-241: In the legends of Figure 1, please mention what device was used to capture a photo.

Line 247-248: Figure 2, A, B,C: Please explain also explain about intensity (T1 weightage or T2). If you are providing data for MRI, I would highly recommend providing the details of MRI operation in methods. What those blue lines are? please indicate.

Table 1: Lines 251-252. The Table has a column with the title "Ref". Under this Ref, there is the text "Current study". I would rather change "Current study" to " see case # 1 or 2 or 3" wherever applicable.

Line 278-279:  As you highlighted PACS gene expression are linked to the binding site of ion channels, and receptors, please discuss how different mechanism such as  Kv channel mutation, the non-synaptic origin of epileptiform activities, Na channel block or Ca-activated K-channel block have wider implication in generation and termination of epileptiform activities. Below are a few studies that can strengthen your current study from a cellular-level perspective.

For reference see
1) PMID: 19893074
2) PMID: 29641306
3) PMID: 26975146

Author Response

Dear Reviewer,

Thank you for the in-depth review and comments. We respond to your comments point-by-point in the following section:

Line 13-25: I would rather provide the full form of the word when it is used for the first time in the manuscript. Eg; PACS2, EIDEE etc.

Answer: Thank you. We have added the full name of PACS2 to the first occurrence of abbreviations.

Still, the sample size (patient) is low n=3! But since it is a human study and it is understandable the research subject is precious and the greater sample size is difficult to achieve!

Answer: Thank you for your understanding.

The introduction and discussion section currently needs more refinement and elaboration to cover depth research bringing various possible mechanisms into context.

Answer: Thank you, PACS2-related EIDEE is rare. We aim to present the clinical and genetic manifestations of the disease. We have elaborated the introduction and discussion as follows and added the suggested references:

Introduction: “Disruptions in ion channel functions could potentially lead to seizures. For example, paroxysmal depolarization shift (PDS) is a hallmark of epileptiform activities at the cellular level in both lower invertebrates and higher mammals, including humans. Investigating simpler invertebrate model systems like leech Retzius nerve cells can help understand the complexity of K+ channels and their roles in epilepsy. Research highlights the non-synaptic nature of PDS, arising from the suppression of calcium-activated K+ channel functions due to voltage-gated calcium channel blockade and unmasking of persistent so-dium current, leading to Na+-dependent PDS in leech Retzius nerve cells [8](2 citation needed). Further research is needed to elucidate the connection between PACS2 mutations, ion channel regulation, and the development of seizures.”

Discussion:  “PACS2, a multifunctional sorting protein, regulates the interplay between the ER and mitochondria, maintaining ER homeostasis [4].  The link between epilepsy and PACS2 remains unclear but may involve mutations affecting ion channel function, leading to seizures. PACS2 gene expression is associated with ion channel binding sites, impacting epileptiform activities [7]. Various mechanisms, such as Kv channel mutations, non-synaptic origin of epileptiform activities, and ion channel blocks, play crucial roles in seizures. Studies demonstrated that Ni(2+)-induced epileptic activity is suppressed by ethanol and magnesium, with Kv1 channels differentially regulating action potentials in etv1 and glt pyramidal neuron. These findings highlight diverse ion channel interactions in epileptiform activity regulation at the cellular level [18-20].”

Line 46-47: Please add another paragraph for the origin, propagation, and termination of seizure at the cellular level. This will give a generalized perspective of epilepsy and does not only limit to the introduction of human cases. For details consult:

1)https://medcraveonline.com/medcrave.org/index.php/MOJAP/article/view/19872/pdf

2) http://www.doiserbia.nb.rs/Article.aspx?id=0354-46641004035P#.ZATTYx_MLrc
With this suggested article, you should be able to define Paroxysmal depolarization shift (PDS) a hallmark of epileptiform activities at the cellular level, and how such features are identical in higher mammals and humans at a cellular level.

Answer: We thank the reviewer's advice. We have added a paragraph into the suggested location to improve readability. 

Introduction: “Disruptions in ion channel functions could potentially lead to seizures. For example, paroxysmal depolarization shift (PDS) is a hallmark of epileptiform activities at the cellular level in both lower invertebrates and higher mammals, including humans. Investigating simpler invertebrate model systems like leech Retzius nerve cells can help understand the complexity of K+ channels and their roles in epilepsy. Research highlights the non-synaptic nature of PDS, arising from the suppression of calcium-activated K+ channel functions due to voltage-gated calcium channel blockade and unmasking of persistent so-dium current, leading to Na+-dependent PDS in leech Retzius nerve cells [8](2 citation needed). Further research is needed to elucidate the connection between PACS2 mutations, ion channel regulation, and the development of seizures.”

Line 240-241: In the legends of Figure 1, please mention what device was used to capture a photo.

Answer: we have added the device used to capture the photos and the permission from the parents to publish the photos. 

Line 247-248: Figure 2, A, B,C: Please explain also explain about intensity (T1 weightage or T2). If you are providing data for MRI, I would highly recommend providing the details of MRI operation in methods. What those blue lines are? please indicate.

Answer: The MRI is a T2-weighted imaging, we have added the information into methods. The blue line is the tegmento-vermian angle. An angle of ≥ 30° indicates the presence of vermian hypoplasia. We have added the explanation to the legend.

Table 1: Lines 251-252. The Table has a column with the title "Ref". Under this Ref, there is the text "Current study". I would rather change "Current study" to " see case # 1 or 2 or 3" wherever applicable.

Answer: We have revised accordingly.

Line 278-279:  As you highlighted PACS gene expression are linked to the binding site of ion channels, and receptors, please discuss how different mechanism such as  Kv channel mutation, the non-synaptic origin of epileptiform activities, Na channel block or Ca-activated K-channel block have wider implication in generation and termination of epileptiform activities. Below are a few studies that can strengthen your current study from a cellular-level perspective.

For reference see
1) PMID: 19893074
2) PMID: 29641306
3) PMID: 26975146

Answer: Thank you for your advice. We have added this to the discussion: 

Discussion:  “PACS2, a multifunctional sorting protein, regulates the interplay between the ER and mitochondria, maintaining ER homeostasis [4].  The link between epilepsy and PACS2 remains unclear but may involve mutations affecting ion channel function, leading to seizures. PACS2 gene expression is associated with ion channel binding sites, impacting epileptiform activities [7]. Various mechanisms, such as Kv channel mutations, non-synaptic origin of epileptiform activities, and ion channel blocks, play crucial roles in seizures. Studies demonstrated that Ni(2+)-induced epileptic activity is suppressed by ethanol and magnesium, with Kv1 channels differentially regulating action potentials in etv1 and glt pyramidal neuron. These findings highlight diverse ion channel interactions in epileptiform activity regulation at the cellular level [18-20].”

Reviewer 2 Report

Dear Authors, 

Extensive work is done by the authors. I really appreciate your enthusiasm. However, I am extremely sorry that the fundamental concept of research writing is wrong here. In the title, you mentioned it as a literature review. The presentation is like a cross-sectional study but you end up like a case series. I want to reject the paper but since you have done good background work and I want to encourage a research attitude among medical people, I suggest you change the study design to case series and make changes in the following study based on these recommendations. Please check my attached PDF for more comments.

chrome-extension://efaidnbmnnnibpcajpcglclefindmkaj/https://cdn-links.lww.com/permalink/jbjsrev/a/jbjsrev_2018_03_28_greysdc_17-00129_sdc2.pdf

chrome-extension://efaidnbmnnnibpcajpcglclefindmkaj/https://jbi.global/sites/default/files/2019-05/JBI_Critical_Appraisal-Checklist_for_Case_Series2017_0.pdf

Author Response

Dear Reviewer,

Thank you for the in-depth review and comments. We changed the title as “Long-term outcome of neonatal seizure with PACS2 mutation: case series and literature review”. We edited the methods based on the case series checklist.

In addition, we respond to your comments point-by-point in the following section:

The complete form of the word when it is used for the first time in the manuscript.

Answer: Thank you. We have added the full name of PACS2 to the first occurrence of abbreviations.

The posterior fossa is a structural term that needs to clarify the imaging findings.

Answer: Thank you. We edited the abstract “Neuroimaging abnormalities were observed in the posterior fossa region, including mega cisterna magna, cerebellar dysplasia, and vermian hypoplasia.”

Abbreviation for Developmental and Intellectual Disabilities as DID or DDID

Answer: Thank you. I kept to the term "Developmental and Intellectual Disabilities." I deleted the abbreviation because there has been no uniform term or abbreviation for the group of patients with developmental disabilities and intellectual disabilities yet.

Materials and Methods, and the headlines should follow the checklist.

Answer: Thank you. We edited the headlines of the materials and methods following the Quality Appraisal Checklist for Case Series Studies as follows:

2.1 study objective

2.2 study design

2.3 Intervention

2.4 Outcome measure

2.5 Literature review

No statistical analysis should be in the headlines.

Answer: Thank you. I deleted the statistical headline of the statistical analysis.

Results should be reported as Quality Appraisal Checklist for Case Series Studies

Answer: Thank you. We edited the headlines of results following the Quality Appraisal Checklist for Case Series Studies as follows:

3.1 Patients

Three cases with a de novo heterozygous missense variant, c.625G>A (p.Glu209Lys), were enrolled in the case series.

3.1.1 Case 1

3.1.2 Case 2

3.1.3 Case 3

3.2 literature review

3.2.1 demographics of 32 cases

3.2.2 Seizure patterns, intervention, and outcomes

3.2.3 Dysmorphism

3.2.4 Neuroimaging features

3.2.5 Clinical neurodevelopmental outcomes

Figure Legends should be under the figures and there was a duplication of table 1 title.

Answer: Thank you. We edited accordingly. The sentence "Informed consent was obtained from the parents of each patient for the use of the images""in the figure legend was moved to the methods.

Figure 2 should mention in detail for figure A, B, and C.

Answer: Thank you. We edited accordingly.

“Figure 2. MRI image features (demonstrated by T2-weighted imaging). (A) Case 1 with mega cisterna magna. (B) Case 2 with vermian hypoplasia (with a tegmento-vermian angle at 32 degrees. (C) Case 3 with mega cisterna magna. The tegmento-vermian angle (illustrated using blue-colored lines) is the angle between the brainstem's dorsal side and the vermis's ventral part. An angle of ≥ 30° indicates the presence of vermian hypoplasia.”

The list of abbreviations should include all abbreviations,  inclduing“3d”.

Answer: Thank you. We checked and used “……d=Day;…m=Month/Months ;…w=Week/Weeks; y=Year/Years” followed by alphabetical order.

Reference should include the author’s name before the reference number.

Answer: Thank you. We edited accordingly.

Discussion with end with your limitation and future suggestion.

Answer: Thank you. We added a paragraph regarding limitation and future suggestion.

“The present investigation is subject to certain limitations that need to be acknowl-edged. Firstly, the inclusion criteria solely comprised individuals diagnosed with epilepsy at Chang Gung Memorial Hospital at the Linkou branch from 2010 to 2022, which may not be a representative sample of the entire population. Consequently, the generalizability of the research findings could be restricted. Moreover, the study design was retrospective, and the data were obtained from the electronic medical records of patients, which might have incomplete or erroneous documentation. Additionally, the sample size of the inves-tigation was limited, with only three cases enrolled. Furthermore, the literature review was confined to English, Chinese, and Japanese languages, which may have constrained the scope of the search results. Future research should conduct a study with a larger and more inclusive sample, utilizing multiple data sources. The literature review should also ex-pand to include publications in other languages and databases, providing a more com-prehensive understanding of PACS2-related EIDEE for clinical practice.”

Need to correct all reference with the surname followed by the abbreviation of initial names.

Answer: Thank you. We edited and checked the references accordingly.
